# Diagnostic accuracy of direct agglutination test, rK39 ELISA and six rapid diagnostic tests among visceral leishmaniasis patients with and without HIV coinfection in Ethiopia

Mekibib Kassa[1], Saïd Abdellati[2], Lieselotte Cnops[2], Bruno C. Bremer Hinckel[3], Arega Yeshanew[1], Wasihun Hailemichael[4], Florian Vogt[2], Wim Adriaensen[2], Pascal Mertens[3], Ermias Diro[1], Johan van Griensven[2], Dorien Van den Bossche[2]*

1 Leishmaniasis Research and Treatment Centre, University of Gondar, Gondar, Ethiopia, 2 Department of Clinical Sciences, Institute of Tropical Medicine, Antwerp, Belgium, 3 Coris BioConcept, Gembloux, Belgium, 4 Department of Immunology and Molecular Biology, Biomedical Sciences, University of Gondar, Ethiopia

* dvandenbossche@itg.be

**Data Availability Statement:** Data cannot be shared publicly because of patient confidentiality.

## Abstract

Diagnosis of a first-time visceral leishmaniasis (VL) infection in Ethiopia is established by use of a rapid diagnostic test (RDT) detecting antibodies against rK39, direct agglutination test (DAT) and microscopy according to the national algorithm. The performance of individual tests and algorithm is variable and depends on several factors, one being HIV status. Limited data are available on the performance of tests in VL-HIV coinfected patients.

Assessment of the performance of DAT (ITM-A), rK39 ELISA (Serion) and six RDT (Onsite Leishmania Ab CTK, Antigen ICT Xinjier, IT Leish Biorad, Kalazar Detect Inbios, rK39 IgG1 Coris, rk28 IgG1 Coris) for the diagnosis of VL was done on a panel of 91 stored serum and plasma samples of 'first-episode' suspected VL patients, with HIV coinfection (n = 51) and without (n = 40). A combined reference standard was used: either positive microscopy on tissue aspirates, or in case of negative microscopy, positive PCR results on the aspirate slide. Additionally, endemic healthy controls (n = 20), non-endemic controls (n = 10) and patients with confirmed malaria infection (n = 10) were tested for specificity evaluation. Sensitivities ranged from 69.2% for DAT (applied cut-off ≥ 1/3200) to 92.2% for the Onsite RDT, whereas specificities ranged from 20.0% for Kalazar Antigen ICT to 100% for IT Leish and rK39 IgG1. Sensitivities from all assays decreased upon stratification according to HIV status but was only significantly different for rK39 Serion ELISA (p-value 0.0084) and the Onsite RDT (p-value 0.0159).

In conclusion, performance of commercially available assays for VL on samples from Northern-Ethiopian patients varied widely with a substantial decrease in sensitivity in the VL-HIV coinfected group. Clear guidelines on minimal performance criteria of individual tests and algorithms are needed, as well as which reference standard should be used to determine the performance.

Data are available from the Institute of Tropical Medicine Institutional Data Access Committee (contact via ITMresearchdataaccess@itg.be) for researchers who meet the criteria for access to confidential data.

**Funding:** This study was part of a larger project supported by the Department of Economy, Science and Innovation of the Flemish Government, and the Belgian Directorate General for Development Cooperation under the ITM-DGDC framework agreement FA-III & FAIV. B.C.B.H. work was supported by funding from the European Union's Horizon 2020 research and innovation programme under the Marie Sklodowska-Curie actions (grant agreement No 642609). The funders had no role in study design, data collection and analysis, decision to publish, or preparation of the manuscript.

**Competing interests:** The authors have declared that no competing interest exist.

## Author summary

In northwest Ethiopia the rates of VL patients coinfected with HIV ranges from 20 to 40%. Limited data are available on the sensitivity and specificity of antibody and antigen detection tests for VL diagnosis in HIV coinfected. Often tests are implemented without prior verification in its specific setting. However, many variables such as regional differences, molecular divergency, age of affected populations and HIV status influence performance and may substantially impact testing algorithms and outcomes.

Several tests were evaluated on a panel of samples obtained in a highly endemic Ethiopian region. rK28 antigen based rapid tests were included as they have previously shown higher sensitivity compared to rk39 based antigen tests. Though this was not confirmed here. Performance was also compared between HIV coinfected and non-HIV infected VL patients. Not all tests equally declined in sensitivity in the HIV coinfected group. Clear guidelines on minimal sensitivity and specificity, with differentiation between HIV and non-HIV infected patients are needed. Even when a national algorithm is available, verification of performance in a specific setting with selected tests and proposed criteria for acceptance remains necessary.

## Introduction

Leishmaniasis presents as a spectrum of diseases and is caused by obligate intracellular protozoa of the genus *Leishmania* which is transmitted by sand flies of the genus *Phlebotomus*. The three main clinical forms of the disease are the cutaneous (CL) form, mucocutaneous (MCL) and the potentially fatal visceral form, visceral leishmaniasis (VL), also called kala-azar [1]. Human VL is present worldwide, but over 90% of cases are found in a few highly-endemic countries (Brazil, Ethiopia, India, Kenya, Somalia, South Sudan and Sudan) [2]. VL is caused by members of the *L. donovani* complex, more specifically the closely related species *L. infantum* and *L. donovani*, the latter being responsible for kala-azar in East Africa.

Patients with kala-azar present with fever, splenomegaly, and weight loss. Clinical diagnosis of VL is inaccurate and can be difficult in endemic settings as several causes of febrile splenomegaly exist, notably malaria [3]. A direct parasitological diagnosis of VL is obtained through microscopy, culture and PCR on tissue aspirates. Obtaining tissue aspirates is invasive and well-trained health professionals are needed to perform this high-risk sampling. However, microscopy lacks sensitivity whereas culture entails weeks to yield a final result and is vulnerable to bacterial contamination. In recent years, molecular techniques have been shown to have the highest diagnostic sensitivity in aspirate or tissue samples, and to be highly specific. However, in resource-limited settings, the use of the PCR remains largely restricted to referral hospitals and well-equipped research centers [4]. Alternative diagnosis by urinary antigen detection or antibody detection through direct agglutination test (DAT), immunofluorescence assays (IFA) and enzyme-linked immunosorbent assays (ELISA) require experienced laboratory technicians, specific laboratory equipment and is time consuming. As many VL patients are poor and live in remote areas, easy and rapid tools for the diagnosis of VL should be selected in accordance with WHO ASSURED criteria (affordable, sensitive, specific, user-friendly, rapid, equipment free and deliverable where required) [5]. Rapid diagnostic tests (RDTs) for *Leishmania* antibody detection have been implemented in endemic settings for over a decade. In combination with clinical symptoms, RDTs provide high diagnostic accuracy in suspected first-time episodes of VL [6, 7]. Performance of commercially available assays differs throughout regions depending on the antigen used, age of the affected population and the

immune status of the patient. Experience with RDT mainly consists of RDTs with antibody detection specifically against recombinant K39 (rK39), a 39-amino acid repeat that is part of a kinesin-related protein whose presence is conserved within the *L. donovani* complex. A limitation of the rK39 based assays is its lower sensitivity in East Africa compared to the Indian subcontinent [1,8, 9]. A new synthetic polyprotein was developed, rk28 which is a fusion molecule of *L. donovani* haspb1, haspb2 and LdK39 tandem repeat regions [10]. Modest improvement in sensitivity of rK28 RDTs over rK39 RDTs was reported in East Africa [11].

VL is highly endemic in both East Africa and the Indian subcontinent but relatively more VL-HIV coinfections are prevalent in East-Africa. Prevalence of coinfection in Northwest Ethiopia varies greatly over different studies. A recent meta-analysis estimated the VL-HIV coinfection prevalence to range from 20.88% to 24.86% [12]. Few studies have assessed the performance of serological assays in the VL-HIV coinfected population in East Africa, but current data suggests a lower sensitivity in VL-HIV coinfected patients [13]. Our objective was to assess the performance of different antibody tests and one antigen test for the diagnosis of new VL cases and compare within and between HIV negative and HIV coinfected patients in an VL- endemic region in Northern Ethiopia.

## Materials and methods

### Ethics statement

Samples from VL suspect cases were leftovers from previous VL studies conducted at the Leishmania Research and Treatment Center (LRTC), Gondar, Ethiopia. Amendments were made to the initial study protocols and approvals were obtained from the Institutional Review Board Antwerp (Belgium), Ethical Committee of the University Hospital Antwerp (Belgium) and the Institutional Review Board of the University of Gondar (Ethiopia) and the Ethiopian National Ethics Committee. According to Institute of Tropical Medicine, Antwerp, Belgium (ITM) policy, approved by the Institutional Review Board, leftover samples of patients presenting at the ITM can be used for test evaluations as long as the patient's identity is not disclosed to third parties and the patient does not explicitly state an objection. For the endemic healthy controls informed written consent was obtained prior to testing with approval of the Institutional Review Board of University of Gondar (Ethiopia).

### Study design

A diagnostic accuracy study on stored samples was conducted to assess the performance of different RDTs, DAT and IgG ELISA for the diagnosis of VL in HIV positive and HIV negative patients. The study was carried out at the LRTCand the ITM. The study design followed the STARD guidelines for presentation of diagnostic studies (S1 Fig) [14].

A panel of biobanked serum and plasma samples from studies performed at LRTC were used for the comparison of the different tests. A confirmed case was defined by either positive microscopy on tissue aspirates, or in case of negative microscopy, positive PCR results on the aspirate slide (reference standard). One hundred samples were selected from suspected VL patients (fever for >2 weeks, splenomegaly and/or abdominal swelling and/or weight loss) with no previous VL history and treatment for VL and stratified by parasite density and HIV status. Out of these, 74 (45 HIV/29 non-HIV) were confirmed VL positive by microscopy on spleen (n = 51), bone marrow (n = 22) and lymph node (n = 1) aspirates. Parasite density grades were 1 (6.8%), 2 (20.3%), 3 (13.5%), 4 (16.2%) 5 (17.6%) or 6 (25.7%). For 26 patients (11 HIV/15 non-HIV) no amastigotes were detected in spleen (n = 18) or bone marrow (n = 8), however 17 were positive by PCR performed on the aspirate slide. Nine VL suspect cases that could not be confirmed by either microscopy or PCR on slide were excluded from

the analysis. This resulted in a total of 91 VL laboratory confirmed VL cases for RDT evaluation, of which 51 came from HIV coinfected patients.

Additionally, 40 samples from non-VL patients were evaluated. Samples from twenty endemic healthy Ethiopian controls, were collected and analyzed with all RDTs and rK39 ELISA, except for the Kalazar antigen ICT. Ten left-over samples from Belgian travelers presenting at the polyclinic of ITM without VL, were selected as non-endemic controls; malaria was excluded by microscopy and RDTs in these patients. Ten malaria positive were included to evaluate cross-reactivity (4 *P. falciparum*, 3 *P. vivax*, 1 *P. ovale*, 1 *P. malariae* and 1 *P. falciparum* and *P. ovale* mixed infection). All control samples tested negative by DAT as a reference and were stored at -80˚C.

## Diagnostic tests

Microscopy was performed on tissue aspirates from either bone marrow, spleen or lymph node. Slides were examined with the 100x objective after being air dried, fixed with methanol and stained with Giemsa 1:10 solution. Parasite density is estimated based on World Health Organization recommended average amastigote density grading of slides [15], grade 1 to 6.

The Direct agglutination test (DAT, Institute of Tropical Medicine—Antwerp (ITM), Belgium) was conducted with a freeze-dried version of the DAT antigen composed of fixed, trypsin-treated and stained promastigotes of *L. donovani*. DAT testing was done as previously described [15]. For result interpretation, two different cut-offs were applied, ≥ 1/800 and ≥ 1/3200, based on the borderline and positive cut-off used in the National Ethiopian guidelines [16].

Leishmania IgG ELISA (Serion Leishmania IgG ELISA, Virion-Serion, Würzburg, Germany) is a quantitative immunoassay which detects IgG antibodies against *Leishmania* spp. by use of the recombinant antigen K39 (rK39) and was performed according to manufacturer's instructions. Results are expressed in U/mL (negative: <10U/mL; borderline: 10–15 U/mL; positive: ≥15 U/mL). Borderline results were considered positive for statistical analysis.

A *Leishmania* antigen detection test (Kalazar Antigen ICT, Xinjier Biotechnology Co, Shanghai, China) was selected for evaluation of presence of *Leishmania* antigen in serum or plasma. IT-Leish (Biorad, USA), Kalazar Detect Rapid test (InBios, USA) and OnSite Leishmania Ab Rapid test (CTK, USA) are commercially available immunochromatographic tests for the qualitative detection of anti-leishmanial antibodies based on reaction with rK39 antigen (anti-rK39) for the first two and with rK28 antigen (anti-rK28) for the latter test. Two prototype tests detecting IgG1 against either rK39 or rK28 (Coris BioConcept, Belgium) were evaluated. These consisted of a cassette with a nitrocellulose membrane sensitized with rK28 (concentration 0.25mg/mL) or rK39 (concentration 0.6mg/mL). Anti-human IgG1 specific antibody labelled with colloidal gold was dried onto the conjugate pad. The control line consisted of goat anti-chicken (GAC) antibodies which was detected by chicken IgY conjugate, impregnated together with the anti-human IgG1 conjugate. 3.5μL of serum or plasma was dispensed on the nitrocellulose membrane, followed by 150μL of buffer in the buffer well. Results were read after 15 minutes. A test was deemed valid if a red control band developed where the control antibody was present, and deemed positive if a second band developed where the rK28 or rK39 had been coated. A test was deemed negative if only the control band developed.

Results were read in sufficient day light and assisted by a standard light source. Photographs were taken from each RDT within the respective allowed time for result reading interpretation. All RDTs were interpreted as negative or, if positive, the intensity of the test line was scored against the control line's intensity (1: weaker than; 2: equal to; 3: stronger than control line). DAT and RDTs were interpreted independently by two lab technicians who were blinded to

the reference test results and clinical information. In case of discrepancies between the technicians, a third person and/or photographs were used as a tiebreaker.

DNA was extracted from Giemsa-stained spleen or bone marrow smear slides using the LEV DNA extraction kits (Promega, The Netherlands) with the Maxwell 16 automate (Promega) according to the manufacturer's instructions with a slightly adapted first step. Briefly, 30 μL of lysis buffer from the kit was dropped onto the smear and the material was scraped off with a sterile Bistouri knife and added to a DNA/RNA free 2 mL tube containing a final volume of 300 μL lysis buffer. After a quick spin, proteinase K was added, and the standard procedure was further followed. The kDNA PCR was performed as described before [17].

## Data management and analysis

Data were recorded on register forms and entered in a Microsoft Excel database (Microsoft Corporation, Redmond, Washington, USA). Statistical analysis was done by use of Analyse-it for Excel (version 5.40.2). Test performance was analyzed by calculating sensitivity and specificity with Wilcoxon 95% confidence intervals for each test. Sensitivities between HIV positive and HIV negative patients were assessed for statistical significance using a two-tailed Fisher-exact test (p<0.05). Pearson Chi square (95% significance level) was used for correlation of results with parasite density. Inter-observer agreement was assessed by determining weighted kappa-values based on the scores given during RDT interpretation and interpreted according to Landis and Koch [18]: 1.00–0.81 excellent, 0.80–0.61 good, 0.60–0.41 moderate, 0.40–0.21 weak and 0.20–0.00 negligible agreement. Also, one weak positive sample was tested five times on different days to evaluate reproducibility.

## Results

### Test performance

No invalid test results were obtained for any of the tests. Performance of all evaluated tests in both HIV and non-HIV VL patients and controls are presented in Table 1. Overall, sensitivities ranged from 69.2% for DAT with cut-off ≥ 1/3200 to 92.2% for the Onsite RDT. Sensitivity was lower in the HIV positive group for all tests, though this was only statistically significant for rK39 Serion ELISA (p = 0.0084) and OnSite RDT (p = 0.0159) for which sensitivity in the non-HIV group reached 100%. DAT titers and rK39 Serion ELISA results according to HIV coinfection are illustrated in Fig 1. For DAT there was no significant difference (p = 0.1856) between proportions (number of patients with a given titer or ratio) of the HIV positive and HIV negative group, opposed to rK39 Serion ELISA (p = 0.0270).

There was no statistically significant correlation between the test result proportions and parasite density for any of the tests except for DAT (Pearson Chi square p-value 0.0419 with cut-off ≥ 1/800 and 0.0019 with cut-off ≥ 1/3200). Also at a cut-off ≥ 1/3200, DAT was positive in 55 out of 74 (74.3%) microscopy positive cases, and only in 8 out of 17 (47.1%) microscopically negative cases (confirmed through PCR on slide).

Overall specificities ranged from 20,0% for Kalazar Antigen ICT to 100.0% for IT Leish, and rK39 IgG1. None of the tests were positive in the 20 endemic healthy controls. Out of 10 non-endemic control samples 1 sample was positive with the Kalazar Detect RDT, 1 other sample for Onsite RDT and 8 samples with the Kalazar Antigen ICT. One out of 10 malaria positive samples tested positive with both rK28 IgG1 and Leishmania IgG ELISA, though still within borderline limits. Two other malaria samples were found positive with the Kalazar Detect RDT and eight with Kalazar Antigen ICT and Onsite RDT.

Agreement between two tests was calculated (Table 2). The highest agreement was noted between rk39 Serion ELISA and IT Leish (96.9%) and the lowest between DAT at cut-off ≥ 1/

**Table 1. Sensitivities and specificities of DAT, rk39 ELISA and six RDTs for VL diagnosis.**

| | company | detection of | antigen | Sensitivity % (95% CIs) | | | Specificity % (95% CIs) | number of false reactives | | |
|---|---|---|---|---|---|---|---|---|---|---|
| | | | | All (n = 91) | HIV (n = 51) | Non-HIV (n = 40) | All (n = 40) | EC (n = 20) | NEC (n = 10) | Malaria (n = 10) |
| **DAT** | | | | | | | | | | |
| ≥800 | ITM | Total antibodies | *L. donovani promastigote* | **75.8 (66.1–83.5)** | 70.6 (57.0–81.3) | 82.5 (68.1–91.3) | **NA** | NA | NA | NA |
| ≥3200 | | | | **69.2 (59.1–77.8)** | 64.7 (51.0–76.4) | 75.0 (59.8–85.8) | | | | |
| **ELISA IgG** | | | | | | | | | | |
| rK39 IgG | Serion | IgG | *Leishmania spp* rK39 | **91.2 (83.6–95.5)** | 84.3 (72.0–91.8) | 100.0* (91.2–100.0) | **97.5 (87.1–99.6)** | 0 | 0 | 1** |
| **RDTs** | | | | | | | | | | |
| IT Leish | BioRad | IgG | rK39 | **91.2 (83.6–95.5)** | 88.2 (76.6–94.5) | 95.0 (83.5–98.6) | **100 (91.2–100)** | 0 | 0 | 0 |
| Kalazar Detect | InBios | IgG | rK39 | **89.0 (80.9–93.9)** | 84.3 (72.0–91.8) | 95.0 (83.5–98.6) | **92.5 (80.1–97.4)** | 0 | 1 | 2& |
| rK39 IgG1 | Coris | IgG1 | rK39 | **76.9 (67.3–84.4)** | 74.5 (61.1–84.5) | 80.0 (65.2–89.5) | **100 (91.2–100)** | 0 | 0 | 0 |
| rK28 IgG1 | Coris | IgG1 | rK28 | **75.8 (66.1–83.5)** | 70.6 (57.0–81.3) | 82.5 (68.1–91.3) | **97.5 (87.1–99.6)** | 0 | 0 | 1$ |
| Kalazar Antigen ICT | Xinjier | antigen | NA | **89.0 (80.9–93.9)** | 88.2 (76.6–94.5) | 90.0 (76.9–96.0) | **20.0 (8.1–41.6)** | ND | 8 | 8! |
| OnSite | CTK | Total antibodies | rK28 | **92.2 (84.8–96.2)** | 86.0 (73.8–93.0) | 100 (91.2–100) | **77.5 (62.5–87.7)** | 0 | 1 | 8μ |

NA: not applicable, ND: not done; EC: endemic healthy controls, NEC: non-endemic controls

* 2 out of 40 samples had values within the borderline range

**1 *P. vivax* infection (borderline rK39 ELISA result)

& 1 *P. falciparum* and 1 *P. malariae* infection

$ 1 *P. vivax* infection

! 2 *P. falciparum*, 3 *P. vivax*, 1 *P. ovale*, 1 *P. malariae* and 1 *P. falciparum* and *P. ovale* mixed infection

μ 3 *P. falciparum*, 3 *P. vivax*, 1 *P. malariae* and 1 *P. falciparum* and *P. ovale* mixed infection

3200 and Kalazar Antigen ICT (56.8%). Median values obtained by rK39 ELISA were compared across qualitative results.

## Inter observer agreement and reproducibility

Overall % agreement between readers in positive and negative test results was 100% for all RDTs, except for rK39 IgG1 (99.2%) and rK28 IgG1 (97.5%). The number of samples for which both readers differed in intensity reading was 3/131 (2.3%) for Onsite RDT and 3/111 (2.7%) Kalazar Antigen ICT, 6/131 (4.6%) for Kalazar Detect, 7/131 (5.3%) for IT-Leish, 16/131 (12.2%) for rK28 IgG1 and 18/131 (13.7%) for rK39 IgG1. Weighted kappa values for between-reader line-intensity readings were calculated for each RDT and all were equal or above 0.89. Test results were reproducible with only one discordance in line intensity over five days for rK39 IgG1 remaining within one category of difference (rated 1 to 2).

## Discussion

With this study we aimed to assess diagnostic accuracy of multiple tests for VL diagnosis in HIV-coinfected Ethiopian patients as data are scarce. Our data have shown that sensitivity

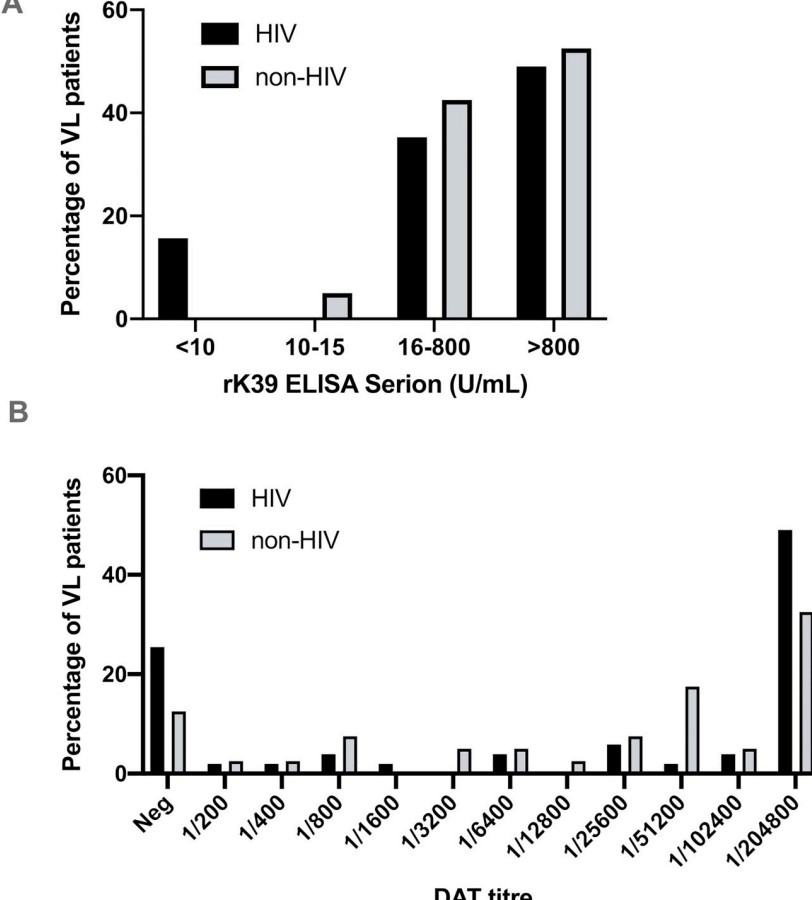

**Fig 1. DAT and rK39 ELISA results stratified by VL-HIV coinfection.** (A) rK39 ELISA Serion U/mL results
compared between VL-HIV co-infected (black) and non-HIV VL patients (light grey) and (B)DAT titer distribution.

decreased in the HIV-VL coinfected group with some tests being more affected than others.
To date, there is a lack of evidence for the minimum sensitivity and specificity of a single test
or algorithm for VL diagnosis. As clinical presentation lacks specificity, and treatment is toxic,
laboratory confirmation is necessary. Previously it was recommended to have a minimum sen-
sitivity of 95% [19] and the test should be able to make a distinction between asymptomatic
patients, treated patients and acute diseased patients. Current Ethiopian national guidelines on

**Table 2. Percentage of concordant results between all tests.**

| % concordant (n = 131) | rK39 ELISA | IT Leish | Kalazar Detect | rK39 IgG1 | rK28 IgG1 | Kalazar Ag ICT | OnSite |
|---|---|---|---|---|---|---|---|
| DAT ≥ 1/800 | 87.8 | 87.8 | 84.7 | 84.0 | 81.7 | 62.2 | 85.4 |
| DAT ≥ 1/3200 | 84 | 84.7 | 82.4 | 82.4 | 79.4 | 56.8 | 75.4 |
| rK39 ELISA Serion | | 96.9 | 95.4 | 90.1 | 87.8 | 69.4 | 96.2 |
| IT Leish | | | 92.4 | 90.1 | 87.8 | 71.2 | 94.6 |
| Kalazar Detect | | | | 90.1 | 84.7 | 71.2 | 91.5 |
| rK39 IgG1 | | | | | 91.6 | 63.1 | 87.7 |
| rK28 IgG1 | | | | | | 62.2 | 86.9 |
| Kalazar Antigen ICT | | | | | | | 71.8 |

the diagnosis of a first-time VL episode start with a non-invasive anti-rK39 RDT upon presentation of clinical symptoms [16]. If positive, the patient should be considered as a new VL case and receive treatment. If negative, a DAT test is performed with a positive cut-off titer of $\geq 1/3200$ and negative $\leq 1/400$. DAT positive patients are referred for treatment, borderline results should be repeated or splenic, bone marrow, lymph node aspiration could be considered if safe. Rarely a DAT negative result would require aspiration. There is no different algorithm for HIV patients, and even with reduced immune response due to HIV infection, sensitivity is still considered acceptable in the national guideline. An algorithm with high sensitivity to rule out or exclude VL infection and a high specificity to avoid these invasive sampling procedures and unnecessary treatment is needed.

Overall, none of the tests evaluated here reach an individual sensitivity of 95% nor sufficiently high specificity. When applying the current algorithm to these results with the most sensitive RDT (Onsite RDT), DAT and aspiration if required, still 6 out of 90 (6.7%) cases (all HIV positive patients) went undetected and 95% sensitivity was not reached either in our study. As per previous study protocol, 5 of these 6 patients underwent splenic or bone marrow aspiration and obtained an aspirate positive (grades 2 to 6) result but they may not have been tested according to the algorithm because of a negative DAT result. Replacing DAT by the second most sensitive RDT (IT-Leish), would result in detection of one additional case. Remarkably, DAT was less sensitive in comparison with previous reported East African results, ranging from 83.5% to 99.1% [11,20–27]. Of note, sensitivities of DAT are often compared against a parasitological diagnosis obtained through microscopy which already lacks sensitivity. This was also shown here as 17 additional cases were afterwards detected through PCR on slides reported negative by microscopy. There are only limited data on the sensitivity of PCR on Giemsa-stained slides compared to microscopy on VL tissue aspirates, however it seems to have improved sensitivity [28, 29]. Assays detecting total antibodies against rK39 all showed a comparable sensitivity ranging from 89.0% to 91.2%. There was no significant difference in this study with sensitivity from the Onsite RDT detecting antibodies against rK28, though previously this has been reported to be higher in East African patients compared to rK39 antigen based RDTs [10, 11]. Presumably the reduced sensitivity of rK39 RDTs in East Africa may be a reflection of molecular diversity of the kinesin gene [30], or by human genetic differences moreover as VL is mainly a pediatric disease in the Indian subcontinent compared to East Africa where all age groups are affected [11]. Not only throughout continents, but also differences in sensitivities within a country may occur, as recently reported by a group from Brazil [31]. Albeit the Onsite RDT exhibits a high sensitivity, a problem implementing this as a screening test is its low specificity in a malaria endemic setting, as 8 out of 10 acute malaria cases reacted false positive (Table 1). Malaria is highly endemic in Ethiopia with an estimated of almost 3 million new cases in 2016 [32]. Prevalence of VL/malaria co-infection in Ethiopia remains largely unknown, but in the Metema hospital in Ethiopia a prevalence of 4.3% has been reported [33]. Aside from coated antigens, other elements influence performance, e.g. the format (ELISA, RDT, lateral-flow vs dipstick. . .), conjugate used, concentrations of components, sample type (serum, plasma or whole blood) and sample volume applied, etc. The Onsite RDT requires a higher sample volume for testing compared to the other RDTs. Dilution of sera has previously been successful in eliminating cross-reactivity against malaria [25], however this may negatively impact sensitivity. An rK39 antigen used in RDT format showed lower sensitivity in Sudan compared to Indian patients, but the same antigen in an ELISA format, results in comparable sensitivities across countries [25]. Here, sensitivity for the anti-rK39 ELISA was comparable to the anti-rk39 RDTs detecting IgG antibodies. None of the RDTs intrinsic quality suffered from invalid results, inter-reader variability or reproducibility problems.

Though these RDTs are implemented in diagnostic settings, they do not distinguish between an acute and treated infection as they detect either IgG or total antibodies. The only antigen detecting test evaluated here did not prove valuable because of its very low specificity. Previously, IgG1 was found to be the predominant antibody present in VL infection and has been explored as a potential biomarker of post-chemotherapeutic relapse [34, 35]. Therefore it was hypothesized that IgG1 detection may also be useful in predicting progression to VL infection among asymptomatic patients which was previously supported by some limited data [35]. Rates of asymptomatic *Leishmania* infection in endemic regions in Ethiopia are estimated to be 10–20% [36] and there is a need for biomarkers to predict progression. Data on the IgG1 titers and sensitivity in VL diagnosis and especially HIV/VL coinfected patients are still sparse [35]. Here, the 2 prototype IgG1 detection antibody tests against rK39 and rK28 antigens showed lower sensitivities compared to total IgG antibody detection when used as a diagnostic assay. Sensitivity in Indian VL patients for the rK39 IgG1 RDT was previously reported to be 94,7%–100% depending on the concentration of antigen used [37]. The lower sensitivity obtained here could be due to different population characteristics between continents. However, samples in the study by Mollett *et al* were selected based on a rK39 total IgG positive test and microscopy at diagnosis, which could contribute to the higher sensitivity they found. The observed results confirms that the IgG1 RDT's are not intended for use as a VL diagnostic assay but as a test of cure or relapse. Moreover these sensitivities may be improved by increasing reagent concentrations, combining rK28 and rK39 antigens or using *Leishmania* lysate as previously described [35].

In northwest Ethiopia, up to 24.% of VL cases are HIV co-infected [12] and this is likely an underestimation as only 17% of VL cases are screened for HIV [38]. Current guidelines recommend HIV screening in VL cases [36]. However, HIV screening should already be performed upon clinical suspicion of VL to guide correct diagnostic algorithms. Sensitivities from all assays decreased upon stratification by HIV status, but the impact of an HIV co-infection was significant for two tests, the rK39 Serion ELISA and the Onsite RDT. Lower sensitivities are a corroboration of previous results [21,31] and likely due to the decrease in CD4$^+$ T-cells- with as a consequence impaired B-cell stimulation and reduced antibody response [39]. Gradoni *et al* [40] suggested that the sequence in which both infections are acquired may impact the seropositivity for VL, unfortunately it is often impossible to retrieve which infection was acquired first. Antibody titers in the rK39 ELISA were significantly lower in the HIV coinfected group, though this was not significant for DAT. In contrast to our results, a meta-analysis proved DAT to be the most effective serological technique in the immunosuppressed by HIV infection [21]. Some tests were 100% sensitive in the non-HIV infected group and could be used to rule out VL infection. However, the current algorithm does not allow to rule out VL among the HIV infected. Agreement of test results between assays were calculated to assess if other algorithms would be of interest or tests could be used in parallel to increase sensitivity and specificity, however with the tests evaluated here this is not possible. Even when combining all antibody detection tests together, still four out of 51 HIV VL coinfected cases would go undiagnosed. Current antibody test cannot reliably rule out VL infection in HIV infected patients. Recently, a test was developed combining the advantages of both DAT and RDTs, with a leishmanial membrane extract used in an RDT format [41]. In Ethiopia this test had 100% sensitivity, but low specificity. It could be a valuable tool to rule out VL in the future, but more data are needed, especially in HIV patients. Other antigens have been explored, like rKLO8 and rKE16 [25,42], but also demonstrated regional differences in performance.

Limitations of this study are (i) it's relatively small sample size influences the variability of results obtained, (ii) samples were analyzed retrospectively and might have undergone multiple freeze-thaw cycles, with a negative impact on sensitivity, however, antibodies are unlikely

to decrease substantially and this would apply to all assays evaluated, (iii) results from patients in control groups were compared against DAT as a reference test as opposed to PCR and/or microscopy on tissue aspirates, which is invasive—this mayintroduce verification bias, (iv) PCR was not systematically performed on all slides; however, microscopy positive results are highly specific and less sensitive compared to PCR and this would unlikely have changed the results, (iii) we have no information on what infection was acquired first, HIV or VL, nor on HIV disease status or HAART, which may influence antibody titers.

Performance of commercial antibody detecting RDTs was variable and sub-optimal. Improvement of sensitivity and specificity with better diagnostic tests is needed. The IgG1-based assays are not appropriate as screening assays due to low sensitivity, but need to be assessed as tools for follow-up of treatment. Assay performance depends on regional differences, associated with parasite divergency and on patient characteristics. Therefore, it is strongly recommended to select the appropriate tests and algorithm to a specific endemic setting and verify performance before implementing it. Separate algorithms for HIV coinfected against non-HIV infected would be valuable. Negative serology does not reliably rule out VL infection in HIV patients. This highlights the need to establish clear guidelines on minimal performance criteria of individual tests and algorithms, but also against which reference these should be compared as parasitological identification through microscopy only has low sensitivity. Large prospective evaluations of both serological and molecular tests taking all these variables into account are required to enable creation of clear guidelines for use of tests in the field.

## Supporting information

**S1 Fig. STARD Flow diagram: Setup and flow of samples throughout the study.**
(TIF)

## Acknowledgments

We want to acknowledge all staff from the Leishmania Research and Treatment Centre, Gondar, Ethiopia for their cooperation on this project. We thank Michael Miles and Tapan Bhattacharya for providing the rK28 antigen for development of the IgG1 rapid test.

## Author Contributions

**Conceptualization:** Lieselotte Cnops, Wim Adriaensen, Johan van Griensven, Dorien Van den Bossche.

**Data curation:** Saïd Abdellati, Lieselotte Cnops, Dorien Van den Bossche.

**Formal analysis:** Mekibib Kassa, Saïd Abdellati, Lieselotte Cnops, Wasihun Hailemichael.

**Funding acquisition:** Ermias Diro, Johan van Griensven.

**Investigation:** Mekibib Kassa, Saïd Abdellati, Lieselotte Cnops, Bruno C. Bremer Hinckel.

**Methodology:** Mekibib Kassa, Lieselotte Cnops, Wasihun Hailemichael, Florian Vogt, Wim Adriaensen, Johan van Griensven, Dorien Van den Bossche.

**Project administration:** Mekibib Kassa, Saïd Abdellati, Dorien Van den Bossche.

**Resources:** Bruno C. Bremer Hinckel, Arega Yeshanew, Wasihun Hailemichael, Wim Adriaensen, Pascal Mertens, Ermias Diro, Johan van Griensven.

**Supervision:** Lieselotte Cnops, Arega Yeshanew, Pascal Mertens, Ermias Diro, Johan van Griensven, Dorien Van den Bossche.

**Validation:** Mekibib Kassa, Saïd Abdellati, Bruno C. Bremer Hinckel, Dorien Van den Bossche.

**Writing – original draft:** Mekibib Kassa, Dorien Van den Bossche.

**Writing – review & editing:** Mekibib Kassa, Saïd Abdellati, Lieselotte Cnops, Bruno C. Bremer Hinckel, Arega Yeshanew, Wasihun Hailemichael, Florian Vogt, Wim Adriaensen, Pascal Mertens, Ermias Diro, Johan van Griensven, Dorien Van den Bossche.

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
