## [Decision Letter · Decision Letter 0]

11 Sep 2020

Dear Ms Van den Bossche,

Thank you very much for submitting your manuscript "Diagnostic accuracy of direct agglutination test, rK39 ELISA and six rapid diagnostic tests among visceral leishmaniasis patients with and without HIV coinfection in Ethiopia" for consideration at PLOS Neglected Tropical Diseases. As with all papers reviewed by the journal, your manuscript was reviewed by members of the editorial board and by several independent reviewers. The reviewers appreciated the attention to an important topic. Based on the reviews, we are likely to accept this manuscript for publication, providing that you modify the manuscript according to the review recommendations. 

Sincerely,

Zvi Bentwich, M.D

Associate Editor

S Madison-Antenucci

Deputy Editor

Reviewer's Responses to Questions

**Key Review Criteria Required for Acceptance?**

**Methods**

-Are the objectives of the study clearly articulated with a clear testable hypothesis stated?

-Is the study design appropriate to address the stated objectives?

-Is the population clearly described and appropriate for the hypothesis being tested?

-Is the sample size sufficient to ensure adequate power to address the hypothesis being tested?

-Were correct statistical analysis used to support conclusions?

-Are there concerns about ethical or regulatory requirements being met?

Reviewer #1: Objectives clearly stated, study design is appropriate, sample size is adequate and statistical analysis is appropriate. No ethical issues identified.

Reviewer #2: Sample size is a concern. It has not been calculated at the outset, and is small. The study has good sensitivity panel, but also poor specificity panel.

Reviewer #3: The objetive is clear and the study design appropriate.

There was an acceptable number of clinical samples available for analysis.

The statistical methods were adequate.

No ethical concerns

**Results**

-Does the analysis presented match the analysis plan?

-Are the results clearly and completely presented?

-Are the figures (Tables, Images) of sufficient quality for clarity?

Reviewer #1: Results clearly presented, including Table and Figure.

Reviewer #2: Some corrections in the figures needed (see separate comment). Diagnostic accuracy has not been determined. See also separate comment.

Reviewer #3: The results were clearly described.

The figure and tables were clear.

**Conclusions**

-Are the conclusions supported by the data presented?

-Are the limitations of analysis clearly described?

-Do the authors discuss how these data can be helpful to advance our understanding of the topic under study?

-Is public health relevance addressed?

Reviewer #1: Conclusions are explicitly presented and shortcomings of the study described in detail. Public health relevance addressed suficiently.

Reviewer #2: It is difficult to know how the results can advance knowledge in the field of diagnostics. The data adds confusion to current data on diagnostic accuracy of VL diagnostics.

Reviewer #3: The conclusions were in line with the results.

The authors discuss how the data suggested that new algorithms for HIV positive or HIV negative need to be considered.

The information has public health relevance for Ethiopia and other countries endemic to VL

**Editorial and Data Presentation Modifications?**

Reviewer #1: Minor revision. Comments summarized in below.

Reviewer #2: There are many editorial comments, please see the separate note.

Reviewer #3: Abstract: It is a little confusing; at the beginning of the abstract, the number of tests need to identify the number of RDTs evaluated in the study. In the lower section of the abstract, the reader doesn't know exactly what tests the authors are referring to as compared with the initial sentence. The abstract has to be self explanatory, not assuming that the reader will read the whole article.

The last sentence used in the abstract and conclusion, is odd and needs to be rephrased "Clear guidelines on minimal performance criteria of individual tests and algorithms, but also against which reference these should be compared are needed."

The term de novo is not commonly used and it could be replaced or explained.

The medical dictionary states:

de novo 

1. Over again from the beginning; anew.

2. Previously undetected. 

3. Previously untreated. 

Line 118, correct this sentence "...no previous VL history and stratified by PARASITE density and HIV status"

How were the DAT cutoff dilutions established?

Table 1. Why was "NA" used in % specificity, etc. for DAT?

Figure 1. The order of graphs was inverted in the legend

**Summary and General Comments**

Reviewer #1: Kassa M and colleagues analyzed the diagnostic accuracy of several assays for visceral leishmaniasis (+ HIV co-infection) in an endemic area in Ethiopia.

Major comments:

1. The investigators noted (page 5, line 98) that VL-HIV coinfection rate in Ethiopia range from 20 to 40%, citing a reference that is relatively old. Nonetheless, a more recent systematic review and meta-analysis demonstrated that overall prevalence of HIV in VL ranges from 20.9% to 24.9% (Mohebali M and Yimam Y. BMC Infect Dis 2020; 20:214. doi:10.1186/s12879-020-4935-x). Thus, this statement needs revisiting. In addition same should be done discussion part (line 333).

2. Methods section (page 6, line 128): What is the DAT result of the ten Belgian travelers (Non-endemic controls)? The reviewer assumes that the results are negative. If so, the authors need to explicitly mention it.

3. Methods section: In page 7, line 141, the authors used two different cut-off titers in interpreting DAT results, namely ≥ 1/800 and ≥ 1/3200, given that DAT is considered as negative if titer is ≤ 1/400, positive at ≥ 1/3200, and borderline at ≥ 1/800. No justification was provided, however, for using such two different cut-offs. Needs explanation. In addition, where do we put/classify those DAT titers with ≥ 1/1600?

4. What are the reasons why borderline rK39-ELISA were considered as positive in the analyses (line 147)? Why not as negative?

5. Line 171: DNA was extracted from Giemsa-stained tissue smear slides. Is there any data known previously the accuracy of the results in using DNA from Giemsa-stained smears? Will this impact on the results as PCR was used in this study as a ‘reference test’ for assessing the sensitivity of the different assays for VL diagnosis employed in this study.

6. Line 212: With respect to Figure 1 on titer proportions of the HIV positive vs. HIV negative VL group – which DAT titer proportions are you discussing? At ≥ 1/800 or ≥ 1/3200? or both? Needs clarifications. In addition, you need to present exact data on proportions (with 95% CI) of rK39-ELISA (line 213) so that the reader can see the differences easily. Having p value only is not suffice.

7. The lower sensitivity (≈77%, all cases) of rK39 IgG1 RDT in Ethiopian VL (line 325) was unexpected when compared with the 94.7% to 100% sensitivity among Indian VL cases. Several reasons were provided by the investigators. One notable difference is the fact that in the Indian study, the ‘reference assay’ used was parasitology only, whereas for the Ethiopian study, it was either parasitology or PCR. Have you tried to dis-aggregated data analysis by ‘reference test’ (74 VL cases by microscopy vs 17 VL cases by PCR), and whether the discrepancies in the sensitivities between the two countries persisted?

Minor comments:

- Please check for proper citations of the references (e.g. reference #s: 11, 19, 21, 23 etc…..)

Reviewer #2: The attached note summarizes this section.

Reviewer #3: Overall, this is a well-written manuscript that has public health relevance for Ethiopia.

It evaluates multiple serological tests available for VL and discusses their utility in the context of algorithms to diagnose the disease in HIV positive or negative individuals.

The Discussion needs to address the role that point of care molecular tests could have for improving diagnostic accuracy. Isothermal amplification tests are (prospectively) a good alternative to complement standard tests, and could be performed at relatively low cost and with minimal equipment and training.

PLOS authors have the option to publish the peer review history of their article (what does this mean?). If published, this will include your full peer review and any attached files.

Reviewer #1: Yes: Dawit Wolday

Reviewer #2: No

Reviewer #3: No
---

## [Editor Report · Decision Letter 1]

9 Nov 2020

Dear Ms Van den Bossche,

We are pleased to inform you that your manuscript 'Diagnostic accuracy of direct agglutination test, rK39 ELISA and six rapid diagnostic tests among visceral leishmaniasis patients with and without HIV coinfection in Ethiopia' has been provisionally accepted for publication in PLOS Neglected Tropical Diseases.

Best regards,

Zvi Bentwich, M.D

Associate Editor

S Madison-Antenucci

Deputy Editor

---

## [Editor Report · Acceptance letter]

9 Dec 2020

Dear Ms Van den Bossche,

We are delighted to inform you that your manuscript, "Diagnostic accuracy of direct agglutination test, rK39 ELISA and six rapid diagnostic tests among visceral leishmaniasis patients with and without HIV coinfection in Ethiopia," has been formally accepted for publication in PLOS Neglected Tropical Diseases.

Best regards,

Shaden Kamhawi

co-Editor-in-Chief

Paul Brindley

co-Editor-in-Chief
